# Pharmacokinetics and Molecular Modeling Indicate nAChRα4-Derived Peptide HAEE Goes through the Blood–Brain Barrier

**DOI:** 10.3390/biom11060909

**Published:** 2021-06-18

**Authors:** Yurii A. Zolotarev, Vladimir A. Mitkevich, Stanislav I. Shram, Alexei A. Adzhubei, Anna P. Tolstova, Oleg B. Talibov, Alexander K. Dadayan, Nikolai F. Myasoyedov, Alexander A. Makarov, Sergey A. Kozin

**Affiliations:** 1Laboratory of Protein Conformational Polymorphism in Health and Disease, Engelhardt Institute of Molecular Biology, Russian Academy of Sciences, 119991 Moscow, Russia; zolya@img.ras.ru (Y.A.Z.); mitkevich@gmail.com (V.A.M.); alexei.adzhubei@gmail.com (A.A.A.); tolstova.anna.pavlovna@gmail.com (A.P.T.); aamakarov@eimb.ru (A.A.M.); 2Department of Physiologically Active Substances Chemistry, Institute of Molecular Genetics of National Research Center «Kurchatov Institute», 123182 Moscow, Russia; shram@img.ras.ru (S.I.S.); dak.img.ras@gmail.com (A.K.D.); nfm@img.ras.ru (N.F.M.); 3Department of Clinical Pharmacology, Faculty of Common Medicine, Evdokimov Moscow State University of Medicine and Dentistry, 127473 Moscow, Russia; oleg.talibov@gmail.com

**Keywords:** Alzheimer’s disease, beta-amyloid, α4β2 nicotinic acetylcholine receptor, peptide drug, blood–brain barrier, receptor-mediated transcytosis, LRP1

## Abstract

One of the treatment strategies for Alzheimer’s disease (AD) is based on the use of pharmacological agents capable of binding to beta-amyloid (Aβ) and blocking its aggregation in the brain. Previously, we found that intravenous administration of the synthetic tetrapeptide Acetyl-His-Ala-Glu-Glu-Amide (HAEE), which is an analogue of the 35–38 region of the α4 subunit of α4β2 nicotinic acetylcholine receptor and specifically binds to the 11–14 site of Aβ, reduced the development of cerebral amyloidogenesis in a mouse model of AD. In the current study on three types of laboratory animals, we determined the biodistribution and tissue localization patterns of HAEE peptide after single intravenous bolus administration. The pharmacokinetic parameters of HAEE were established using uniformly tritium-labeled HAEE. Pharmacokinetic data provided evidence that HAEE goes through the blood–brain barrier. Based on molecular modeling, a role of LRP1 in receptor-mediated transcytosis of HAEE was proposed. Altogether, the results obtained indicate that the anti-amyloid effect of HAEE, previously found in a mouse model of AD, most likely occurs due to its interaction with Aβ species directly in the brain.

## 1. Introduction

Alzheimer’s disease (AD) is characterized by aberrant interactions of beta-amyloid (Aβ) with α4β2 nicotinic acetylcholine receptors [1]. Such interactions can play a pivotal role in the pathogenesis of AD by initiation of the formation of both neurotoxic Aβ oligomers in biological fluids of the body and extracellular insoluble Aβ aggregates in amyloid plaques characteristic of AD [2]. It was previously shown that the Aβ site 11-EVHH-14 [3,4] and the site 35-HAEE-38 of the α4 subunit of the α4β2 subtype of nicotinic acetylcholine receptors (α4β2 nAChR) [5] participate in the Aβ-α4β2 nAChR interaction interface. Synthetic analogs of these sites stabilize the monomeric state of Aβ in vitro and are considered as potential anti-amyloid agents for the treatment of AD [6,7]. Earlier, in transgenic mice used as a model of AD, it was shown that the synthetic tetrapeptide Acetyl-His-Ala-Glu-Glu-Amide (HAEE), when administered intravenously, significantly suppressed the development of cerebral amyloidogenesis [8]. It was suggested that due to specific binding to the EVHH site of the Aβ molecule, the HAEE peptide can destroy both soluble neurotoxic oligomers of Aβ in the bloodstream and aggregates of Aβ at the α4β2 nAChR locations in the brain [7]. In the first case, the HAEE tetrapeptide must be sufficiently stable in the bloodstream; in the second, it must cross the blood–brain barrier (BBB).

In the current study, various aspects of the pharmacokinetics of the HAEE peptide and its distribution in animal tissues were studied after a single bolus injection of tritiated HAEE into the systemic circulation. A non-canonical form of pharmacokinetic curves was found, characterized by the presence of a pronounced maximum at the beginning of the curve, which indicates an atypical behavior of this peptide in the body. This dependence can be explained by interactions of HAEE with blood plasma proteins and/or with receptors located on the endothelial surface. The results of pharmacokinetic studies indicate a rapid clearance of HAEE from the bloodstream and the accumulation of HAEE in the brain. The distribution of charged amino acid residues in the HAEE peptide is the same as that in the C-terminal site of the KTEE of the angiopep-2 peptide [9], which efficiently passes from the bloodstream through the BBB via the receptor-mediated transcytosis (RMP) mechanism due to specific binding to the “barrel” domain of the LRP1 receptor [10]. Using molecular modeling methods, we demonstrated the ability of HAEE to bind to LRP1 sites responsible for angiopep-2 binding. In conjunction with our previous data on the anti-amyloid effect of HAEE in the mouse model of AD [8], the results of this study support the use of HAEE for the targeted destruction of Aβ aggregates directly in the brain tissues.

## 2. Materials and Methods

### 2.1. Animals

The experimental animals used in this study were 24 healthy adult male Wistar rats, each 12 weeks old, weighing 350 ± 30 g; 36 male mice of the C57Bl/6 line, each 6–7 weeks old, weighing 25 ± 2 g; and 3 male Chinchilla rabbits, each 12 weeks old, weighing 2000 ± 200 g. The animals were randomly assigned to groups (Table 1).

The conditions for keeping the animals corresponded to the current sanitary rules for the arrangement, equipment and maintenance of experimental biological clinics. The standard laboratory diet was in accordance with current regulations. Rats and mice were kept in groups of four to a cage, with free access to water and food at a temperature of 21 °C and constant daylight of 14 h (from 8 h to 22 h). The environmental conditions (temperature, humidity, illumination, ventilation, bedding composition) corresponded to the requirements for keeping laboratory animals. Laboratory animals of specific pathogen-free (SPF) status were delivered from the Laboratory Animals Nursery of a branch of the Shemyakin and Ovchinnikov Institute of Bioorganic Chemistry of the Russian Academy of Sciences in Pushchino (Moscow region, Puschino, Russia), which is internationally accredited by AAALAC. All procedures were performed according to protocols approved by the Institutional Review Board of the Institute of Molecular Genetics of the ”Kurchatov Institute” National Research Center. Euthanasia of rabbits and rats was carried out by the introduction of an overdose of chloral hydrate, and euthanasia of mice by the displacement of the cervical vertebrae.

### 2.2. Preparation of Tritium-Labeled HAEE

The synthetic peptide Acetyl-HAEE-Amide (HAEE) with acetyl and amide protecting groups at the *N*- and *C*-termini respectively, of chromatographic purity above 98%, was purchased from Synthon-Lab (St. Petersburg, Russia). The tritium-labeled HAEE peptide ([^3^H]HAEE) was obtained using the reaction of high-temperature solid-state catalytic isotope exchange (HTSIE) [11] with gaseous tritium, at a pressure of 250 Torr and a temperature of 175 °C, in a solid mixture formed by the HAEE peptide supported on calcium carbonate and the highly dispersed catalyst 5% palladium-on-calcium carbonate. The [^3^H]HAEE peptide was desorbed with 20% aqueous ethanol. To remove labile tritium, the [^3^H]HAEE peptide was additionally dissolved twice in 20% aqueous ethanol and evaporated to dryness. The [^3^H]HAEE peptide was purified by HPLC on a Kromasil C18 column, 4 × 150 mm in methanol gradient, in the presence of 0.1% heptafluorobutyric acid. The solution containing [^3^H]HAEE was evaporated, the resulting residue was dissolved in ethanol and the radioactive concentration was adjusted to 1 mCi/mL. The product was then analyzed using radio chromatography. As a result, the peptide [^3^H]HAEE was obtained with molar radioactivity of 40 Ci/mmol and a radiochemical purity of 98%. To obtain [^3^H]HAEE preparations intended for pharmacokinetic studies, an alcoholic solution of the [^3^H]HAEE peptide was evaporated to dryness under reduced pressure and dissolved in physiological saline solution containing a calculated amount of unlabeled HAEE. HPLC profiles of [^3^H]HAEE and HAEE were completely identical.

### 2.3. Administration of [^3^H]HAEE to Animals and Blood Sampling

In experiments on rabbits (Table 1, experimental group 1), peptide administration and blood sampling were performed through the large ear veins. To restrict movement, the rabbit was placed in a restrainer with a head hole. Into the right ear, through the large ear vein, 4000 IU of heparin (FSUE “Moscow Endocrine Plant”, Moscow, Russia) in 0.8 mL of saline solution was injected using an insulin syringe, and after 5 min a Flexicath G22 intravenous catheter (Medica LLC, Ivanovo, Russia) was installed into the large ear vein of the left ear for subsequent blood sampling.

A 0.5 mL of the mixture of HAEE and [^3^H]HAEE (2000 µCi, at a dose of 120 µg/kg) was injected with an insulin syringe into the right ear vein for 10–15 s, and after 2, 4, 6, 10, 20, 30, 60, 90 and 120 min, approximately 0.5 mL of venous blood was collected through the catheter and placed into pre-weighed plastic tubes. Immediately after collection, blood tubes were frozen in liquid nitrogen and weighed.

In experiments on rats (Table 1, experimental groups 2–5), intravenous (i.v.) administration of the peptide and blood sampling were performed through the jugular veins. For this, in rats anesthetized with a mixture of ketamine (91 mg/kg) and xylazine (9.1 mg/kg), on the ventral side, in the area adjacent to the forelimbs and neck, two longitudinal skin incisions about 2.5 cm long were made on the right and left, and the jugular veins were dissected. An intravenous Flexicath G24 catheter (for heparin and peptide administration) (Medica LLC, Ivanovo, Russia) was installed in the left jugular vein and Flexicath G22 (for blood collection) in the right jugular vein. To prevent blood coagulation, 80 μL of heparin (5000 I.U./mL) was injected through the Flexicath G24 port before the injection of the peptide, and after 5 min a mixture of HAEE and [^3^H]HAEE (400 μCi, in doses of 50, 300 or 900 μg/kg) in 200 μL of saline solution was injected for 10–15 s. Following that, after 2, 4, 6, 10, 20 and 30 min, approximately 0.3 mL of venous blood was collected into pre-weighed plastic tubes. After collection, the blood tubes were immediately frozen in liquid nitrogen and weighed. Before the pharmacokinetic study, rats in experimental group 5 were injected daily with i.p. peptide HAEE at a dose of 300 μg/kg for 28 days.

In the experiments on mice (Table 1, experimental group 6), a mixture of HAEE and [^3^H]HAEE (50 µCi, at a dose of 300 µg/kg) in 200 µL of saline solution was injected intraperitoneally. The animals were decapitated at the end of the period specified by the protocol; blood and studied tissues were collected in weighed plastic tubes, frozen with liquid nitrogen and weighed. When calculating the volumetric concentrations of the total amount of the HAEE peptide (labeled, unlabeled or their mixture) in the blood, the density of the blood was taken equal to 1.056 g/mL [12].

### 2.4. HAEE Assay in Tissues Specimens

Frozen and weighed tissue samples in plastic tubes were freeze-dried for 48 h. The freeze-dried tissue samples were heated at 65 °C for 30 min, after which they were homogenized, and the peptide fraction was extracted with 90% aqueous acetonitrile containing 1% TFA [13]. To facilitate HPLC analysis of radioactively labeled HAEE and normalize the loss of the peptide during extraction, 10 μg of HAEE was added to the extraction solution for each sample. After centrifugation, the solution containing HAEE and blood components was dried under reduced pressure, stripped with methanol and re-centrifuged. The resulting solution containing a mixture of HAEE and [^3^H]HAEE was subjected to evaporation under reduced pressure, re-extraction with 0.1% aqueous TFA solution and subsequent centrifugation. Quantitative analysis of the HAEE peptide was performed using HPLC on a Kromasil C18 column, 4 × 150 mm (Nouryon-Separation Products, Bohus, Sweden) in methanol concentration gradient (12–26% for 13 min), in 0.1% HFBA (heptafluorobutyric acid). The fraction containing the HAEE peptide was collected and its [^3^H]HAEE content was determined by liquid scintillation counting. The obtained radioactivity value was normalized to the area of the HAEE peak in the chromatogram (UV absorbance at 220 nm).

### 2.5. Analysis of Pharmacokinetics Data

To calculate the pharmacokinetic parameters, we used the values of HAEE peptide concentrations in blood and tissues averaged over three (rabbit) or six (rats and mice) animals. All quantitative values are given as means ± standard deviation. The calculation of pharmacokinetic parameters was performed by combining the model and non-model approaches using the software Borgia 1.03 (NPP Nauka Plus, Rovno, Ukraine) and SigmaPlot 11.0 (Systat Software, San Jose, CA, USA).

### 2.6. Molecular Modeling

Three-dimensional model structures for the membrane protein LRP1 were found in the ModBase database [14]. We also used the I-Tasser server (accessed on 13 September 2020) [15] to obtain a model of a larger section of the protein. The human LRP1 amino acid sequence, which differs from the murine by substitution of Thr-434 for alanine, was sent to the I-Tasser server. We believed that such a substitution would not affect the predicted protein structure, since such accuracy was beyond the capabilities of the method. We additionally simulated these structures with 50 ns molecular dynamics (MD) in water with a NaCl concentration of 0.115 M in the GROMACS software environment (version 2020.2) [16]. The HAEE tetrapeptide with protected ends (CH_3_CO- and NH_2_- groups were added to the *N*- and *C*-termini, respectively) was built manually and MD-simulated for 50 ns in water with a salt concentration of 0.115 M NaCl in the GROMACS software environment. Angiopep-2 was constructed by expert modeling and equilibrated by MD for 20 ns in water with a salt concentration of 0.115 M NaCl in the GROMACS software environment. Angiopep-2 in all models had a neutral *C*-terminus. 

Targeted docking was carried out using HADDOCK 2.4 server [17]. The docking-involved models had been preliminarily simulated by the MD under experimental conditions. Global docking was done with a locally installed docking program. In total, 30 variants of docking models (structures of complexes) were obtained. The models were then processed with the QASDOM server developed by us [18] (last update 5 June 2018).

The following molecular dynamics protocol was used for all models. Modeling (structure relaxation) was carried out in the GROMACS environment. All models were first subjected to energy minimization sequentially, using the steepest descent, and then conjugated gradients algorithms until a local minimum was reached. Subsequently, a two-stage equilibration of the system was carried out in NVT (the number of particles, volume and temperature were constant) and NPT (the number of particles, pressure and temperature were constant) ensembles of 100 ps, respectively. In the simulation, the Ewald summation algorithm was used and the restrictions on the motion of atoms were set using the LINCS algorithm. The cutoff radii of the Coulomb and Van der Waals potentials were 1.0 nm. The time step was 0.2 fs. All systems included periodic boundary conditions. Water and ions were modeled explicitly and for water the TIP3 model was used.

## 3. Results

### 3.1. In Vitro Stability Assay of Synthetic HAEE

An in vitro assay of the unlabeled peptide stability was performed as described in [19]. Briefly, HAEE was prepared as a 1 mM solution in phosphate-buffered saline and 20 µL of the peptide solution was diluted in 80 µL of human plasma (freshly taken). The solution was incubated at 37 °C for different times, and the reaction was stopped by adding the Roche cOmplete Protease Inhibitor Cocktail (Roche Molecular Biochemicals, Mannheim, Germany). The bulk of the plasma proteins (but none of the peptide) was precipitated in cold methanol (1:4 (*v*/*v*) mixture/MeOH) for 1 h at 20 °C. The precipitated proteins were pelleted by centrifugation at 10,000× *g* for 10 min at 4 °C. The supernatant containing the peptide was concentrated five times under vacuum and separated by RP-HPLC. The area of the peak (UV absorbance at 205 nm) corresponding to the intact peptide was measured and compared with an equivalent sample incubated in phosphate-buffered saline. The peptide was very stable in human plasma, showing no significant degradation within 24 h of incubation at 37 °C (data not shown).

### 3.2. Pharmacokinetic Parameters of the HAEE Peptide after Intravenous Bolus Administration to Rabbits and Rats

Most of the short synthetic peptides are characterized by a short lifetime in the body (the elimination half-life range is from several minutes to tens of minutes when introduced into the central bloodstream), which is caused by their intensive proteolytic degradation and active excretion, mainly by renal clearance [20]. However, in the case of HAEE, we are dealing with a peptide protected at the *N*- and *C*-termini, which contributes to an increase in its resistance to peptidases. Preliminary studies (Section 3.1) showed that HAEE demonstrates very high stability during prolonged incubation in human blood plasma, which may be associated with its binding to plasma proteins. Taking this into account, to determine the pharmacokinetic characteristics of HAEE after systemic administration to rabbits, we analyzed the peptide content in the blood for a long period of time, from 2 to 120 min. The peptide was administered i.v. at a dose of 120 μg/kg. The results of analysis of the peptide concentration in the rabbit blood (Figure 1) were quite unexpected. The form of the pharmacokinetic curve is atypical for the i.v. introduction of drugs, specifically the presence of a pronounced maximum of the peptide concentration in blood 4 min after its administration (Figure 1). At the same time, in the time range 4–120 min, the concentration of the peptide in blood decreases monotonically with time, and the experimental values are best approximated by bi-exponential decay equation (Equation (1)):Ct = A · exp(−α·t) + B · exp(−β·t)(1)
where Ct is the concentration of a pharmacological substance in the blood at time t; and A, B, α and β are hybrid constants (macro-constants). Application of a simpler mathematical model, the equation of mono-exponential decay, led to less adequate results in terms of approximation with the experimental values. Thus, the obtained pharmacokinetic dependence with a single i.v. bolus administration of HAEE to rabbits (Figure 1A,B) can be divided into two components: (1) “non-canonical”, in the time range of 0–4 min, characterized by an increase in the concentration of the peptide by minute 4, and (2) “canonical”, in the time range 4–120 min, characterized by a concentration–time relationship typical for a two-compartment model.

The nature of the dependence revealed above led to the usage of a combination of model and non-model approaches to calculate pharmacokinetic parameters (Table 2). To calculate the correct AUC_(0–∞)_ values, the AUC values were analyzed separately in the 0–4 min interval using the trapezoidal method. It was shown that the pharmacokinetics of HAEE is characterized by a short lifetime of the peptide in the blood T_1/2(el)_ (−20 min, MRT –29 min) and its rapid excretion from the body (Cl_T_—100 mL/min).

In a similar study in rats (i.v. bolus administration of HAEE in doses of 50, 300 and 900 µg/kg), the “non-canonical” pharmacokinetic pattern was fully reproduced (Figure 2A,B). In this case, the process of excretion of the peptide from the body was characterized by approximately the same values of indicators as in the rabbit: the T_1/2(el)_ values varied within 22–25 min; MRT, 32–36 min; and the Cl_T_ value was 6.7–13 mL/min (Table 2).

### 3.3. Examining the Hypothesis of the HAEE Pharmacokinetics Linearity

Confirmation of the linear nature of the pharmacokinetic parameters associated with the concentration of the substance indicated the absence of saturation (or depletion) of the processes involved in the distribution of the test substance in the body, its excretion and biodegradation. It also helped to adjust the dose of the substance to achieve definitive concentrations in blood and tissues. To test the hypothesis of linearity of the HAEE pharmacokinetics, concentration of the peptide in the blood was determined after a single i.v. bolus administration of HAEE to rats in doses of 50, 300 and 900 µg/kg. In all three cases, a similar pattern of peptide concentration versus time was observed, featuring an increase in the peptide concentration in blood from 2 to 4 min and a subsequent gradual decrease in concentration (Figure 2A,B).

The values of pharmacokinetic parameters calculated from the obtained experimental data are summarized in Table 2. To test the hypothesis of linearity, analysis of the following four dose-dependent (by definition) pharmacokinetic parameters was performed: AUC_(0–30)_, AUC_(0–__∞)_, C_max_ and B (a macro-constant in the bi-exponential decay equation) (Equation (1)). Calculations showed that all the above pharmacokinetic parameters were directly proportional to the applied dose of the peptide (Figure 2C). The values of the Pearson correlation coefficient between the peptide dose and the calculated values of AUC_(0–30)_, AUC_(0–__∞)_, C_max_ and B were 1.000 (*p* = 0.005), 0.997 (*p* = 0.052), 1.000 (*p* < 0.001)) and 1.000 (*p* = 0.007), respectively. At the same time, the values of time-dependent parameters (T_max_, MRT, K_el_, T_1/2(el)_) and volume-dependent parameters (Cl_T_, V_d(s)_, V_d(__β)_, V_d(extrap)_), obtained for the examined doses of the peptide did not differ significantly from each other (Table 2). This indicates that only the pharmacokinetic parameters linked to the concentration of the substance show a linear dependence when the dose of the peptide is varied in the range from 50 to 900 μg/kg.

### 3.4. Pharmacokinetics of HAEE after Chronic Administration of the Peptide to Rats

Long-term “course” administration of the HAEE peptide can theoretically lead to a change in its pharmacokinetics, which may be associated with its effect on metabolic processes, the toxic effects on organs involved in the excretion of substance from the body, and with the regulation of the expression and activity of enzymes involved in its metabolism. To study possible manifestations of such effects, we analyzed the changes in HAEE content in the blood for the time range 2–30 min for the control (intact rats, *n* = 6) and experimental (rats that were i.v. injected with HAEE daily for 28 days with a dose of 300 μg/kg, *n* = 6) groups of animals (data not shown). Based on these data, the main pharmacokinetic parameters of HAEE were calculated for each group (Table 2). The values of these parameters in both groups were practically the same. Thus, chronic administration of the HAEE peptide does not cause its accumulation in rats and does not affect the nature of its pharmacokinetics.

### 3.5. Distribution of HAEE in Mouse Tissues

Within the study, bioavailability of HAEE for several organs and tissues was analyzed with i.p. bolus administration to mice at a dose of 300 μg/kg. The quantity of HAEE in the blood, brain, kidneys, liver, heart and omentum was determined at 2, 4, 6, 10, 15 and 20 min after peptide administration. For all tissues, similar changes in the concentrations of HAEE over time were observed (Figure 3A–C). These were characterized by a bimodal relationship with peaks at 4 and 10–15 min. The ratios of peptide concentrations in these time intervals in the kidneys, liver and omentum were approximately equal, while in the blood the 4 min mode was significantly higher than the 10–15 min mode, and in the brain, in contrast, the 10–15-min mode was slightly higher than the 4 min mode.

Analysis of the tissue bioavailability (the ratio of AUC for tissues and blood) showed that the highest bioavailability of HAEE was characteristic of kidneys, where the peptide concentration in the entire investigated time range was higher than in the blood (Table 3). In the liver and heart, comparable amounts of the peptide were found, but significantly lower than in the blood. Tissue bioavailability of HAEE for the brain was about 3%, which is typical for most pharmacological substances. However, it seems to us very significant that an increase in the peptide concentration was observed in the brain over time (10–15 min), even against the background of a decrease in its concentration in the blood (Figure 3C). This may indicate accumulation of HAEE in the brain parenchyma due to its active transfer through the BBB and further binding and retention at the affinity sites of neuronal protein targets.

### 3.6. Molecular Modeling Results

Molecular modeling was used to test our hypothesis that HAEE, like the angiopep-2 peptide, binds to the LRP1 receptor on the luminal surface of the endothelium of neurovascular vessels and passes through the BBB by the mechanism of receptor-mediated transcytosis. LRP1 is a large membrane protein with an extramembrane part of about 4500 a.a., and its crystal structure is unknown. The mouse LRP1 region containing the site that interested us, 281-HHVE-284, was taken for modeling. In the Uniprot sequence database, this region is adjacent to the functional domain of the LDL-receptor class B1 (293–335 a.a.), for which the standard beta-propeller structure is known. The beta-propeller structure was also obtained by crystallography for PDB:1IJQ. Using it as a template, we built models of the human and mouse LRP1 beta-propeller domains that were of interest for this study, including the HHVE regions. These models were taken from the MODBASE database of modeled structures through the method of automatic comparative modeling by the ModPipe program (as a result of regularly conducted searches for homologous sequences/templates in the known three-dimensional structures) and were also built by us using the I-Tasser and Phyre servers.

The final structures for human and mouse LRP1 were almost identical to each other after equilibration of the structures by molecular dynamics, since they have a large region of identical sequence (in our models, the a.a. 200–500 regions coincided almost completely, differing in one amino acid residue). The main difference in 3D structures was the position of the negatively charged region a.a. 298–303, consisting of the DDIDD residues, located in relative proximity to the HHVE site. In the structure of the human LRP1 model, this area was located slightly closer to the HHVE site and was less screened, which could have affected the MD simulation results. However, the overall domain structure, as well as the location of the HHVE site and its residues remained practically unchanged in models obtained using different modeling servers and for different organisms (RMSD after alignment was 4.040 Å for the four structures of the mouse and human beta-propeller LRP1 domains).

We selected a model with a larger coverage of the LRP1 sequence, a.a. 204–516 from MODBASE, and a model for the region of a.a. 182–541 modeled using the I-Tasser server, and performed equilibration of the structures with MD in water environment with NaCl. After 50 ns of molecular dynamics, we were convinced that the structures were stable. However, the structures after MD differed from each other and the RMSD was 5.88 Å. At the next stage of modeling, the HAEE tetrapeptide with protected termini was added to the system. The peptide was placed using expert modeling in parallel to the HHVE site and in close proximity to it. The MD cycle of this system was repeated for 50 ns. The NaCl salt was added to water in each system at a concentration of 0.115 mM, since we believed that the contact occurs in the intercellular space. After 50 ns, HAEE was bound to HHVE, but few hydrogen bonds were formed (Figure 4). Since we assumed an ion-complementary interaction between the HAEE and HHVE sites with the formation of bonds between histidine residues and glutamates, in the next round of modeling we protonated His1 in HAEE, and His281 and His282 in LRP1. Using expert modeling, the HAEE tetrapeptide was positioned relative to the HHVE site in such a way that the formation of three His-Glu hydrogen bonds was possible. Then, the resulting systems were equilibrated by molecular dynamics for 50 ns. Figure 4 shows the structure after 50 ns MD where His1 and Glu4 from HAEE interact with Glu284 and His282 in LRP1 respectively.

To assess the availability and preference of the HHVE as a binding site, we performed global docking of the protonated HAEE tetrapeptide to the LRP1 models from the I-Tasser server and MODBASE. The main binding regions are shown in Figure 5. As shown by docking, in the LRP1 model from the I-Tasser server, the binding site is adjacent to HHVE and therefore many HAEE molecules interact with HHVE, but most of the contact falls in the LRP1 Arg229-Gln230 region, which strongly bind glutamates. In the LRP1 model from ModBase, this region is also accessible, but the tetrapeptide potential binding areas are more evenly distributed over the protein surface.

To assess the role of the HHVE site for binding to the receptor and transfer of the HAEE peptide across the BBB, we simulated the binding of another peptide to LRP1 that is known to penetrate the BBB in this way [10]. This peptide was an angiopep-2 with the sequence TFFYGGSRGKRNNFKTEEY-OH [9]. Its C-terminus contains a KTEE site, which is complementary to the HHVE receptor site. The structure of angiopep-2 is unknown; therefore, using expert modeling, we constructed the model of the peptide with a neutral C-terminus. Targeted docking of this peptide to the HHVE site in the previously created model of the LRP1 beta-propeller domain was carried out using the HADDOCK server. The KTEE site in the angiopep-2 peptide was designated to interact with HHVE. Based on the docking results, two models were selected with many hydrogen bonds between the KTEE sites in the angiopep-2 peptide and HHVE from LRP1. These models were then equilibrated by MD simulation for 50 ns and 30 ns, respectively. In both cases, angiopep-2 retained the bonds formed during the docking process, and the structures remained stable (Figure 6). This result points to a HHVE region at the receptor as a possible site responsible for transfer of the HAEE peptide across the BBB.

## 4. Discussion

Protein–protein interactions (PPIs) are the foundation of essentially all cellular process. Peptides and small molecules that interfere with PPIs are thus in high demand as therapeutic agents in pharmaceutical industries due to their potential to modulate disease-associated protein interactions [21]. Neurodegenerative diseases are typically caused by abnormal aggregations of proteins or peptides, and the depositions of these aggregates in or on neurons disrupt signaling and eventually kill neurons. In recent years, research on short peptides has advanced tremendously [22]. Application of therapeutic peptides for the treatment of Alzheimer’s disease is also widely discussed [23], but the question of the ability of such peptides to penetrate the BBB persists [24].

It can be assumed that Aβ deposition induces degeneration of cholinergic terminals [25], especially at the locations of α4β2 nAChRs [26,27] and α7 nAChRs [28]. Many studies support the notion that Aβ can physically interact with α4β2 nAChRs and α7 nAChRs in various model systems [29,30,31,32]. Since Aβ accumulates in the brain regions enriched in α4β2 nAChRs and α7 nAChRs, the selective vulnerability of the hippocampus to Aβ toxicity can be associated with the high-affinity interaction between Aβ and these nAChRs [33,34,35,36]. The Aβ–α4β2 nAChR interaction interface is formed by the 11-EVHH-14 and 35-HAEE-38 sites of Aβ and the α4 subunit of α4β2 nAChR, respectively [5]. The 11-EVHH-14 region of Aβ also plays a critical role in Aβ binding to α7 nAChRs; however, the exact interface of the Aβ-α7 nAChR complex is unknown [4,6,29,37,38]. The 11-EVHH-14 region has a relatively rigid backbone conformation in soluble Aβ monomers [39,40] and zinc-bound dimers [41]. This site corresponds to the β-strand β2 from the *N*-terminal arch of the Aβ amyloid fibrils purified from Alzheimer’s brain tissue and is solvent-exposed and accessible for interactions with external molecules [42]. Thus, molecular agents binding to the 11-EVHH-14 region of Aβ can modulate interactions between Aβ (in soluble or aggregated states) and α4β2- and α7-containing nAChRs.

The tetrapeptide HAEE had been proposed as one of the candidate molecules for inhibiting cerebral amyloidogenesis in AD [7]. This peptide is a synthetic analogue of the 35-His-Ala-Glu-Glu-38 (35-HAEE-38) fragment of the extracellular *N*-terminal domain of subunit α4 of the α4β2 nAChR. This fragment (conservative for humans, mice and chicken) forms the interface for the interaction of α4β2 nAChR with Aβ [5]. It has been previously shown by molecular modeling methods that this interaction is stabilized by ion bridges between the complementary charged side groups of the H/E and E/H amino acid pairs [5]. The loss of cognitive functions in patients at the middle stages of AD is accompanied by a significant reduction in the availability (according to positron emission tomography data) of the α4β2 nAChRs [1], most likely due to the aggregation of Aβ on these receptors. Accordingly, it had been proposed to use the HAEE peptide as a competitive inhibitor of aberrant interactions of Aβ with α4β2 nAChR [3]. It has been previously found that intravenous injections of the HAEE peptide dramatically slowed down the development of cerebral amyloidogenesis in B6C3-Tg (APPswe, PSEN1-dE9) 85Dbo/j mice, which are used as a recognized animal model of AD. As a result, the average number of amyloid plaques in a brain section decreased from 14.2 ± 3.1 (for control animals) to 5.8 ± 2.1 (for animals subjected to therapeutic effects) [8]. It has recently been shown that amyloid aggregates formed upon contact with α4β2 nAChR in model oocytes, blocking the normal function of receptors, are destroyed by the effect of exogenous HAEE molecules [5].

In this work, in experiments on rabbits of the Chinchilla breed, Wistar rats and mice of the C57Bl/6 line (Table 1), an extended study of the pharmacokinetic properties of HAEE was carried out using radioactively labeled preparations of the HAEE peptide. The study included determination of the values of the main pharmacokinetic parameters, analysis of the effect on these indicators of preliminary long-term administration of unlabeled peptide, testing for the linearity of dose-dependent parameters and analysis of the bioavailability of HAEE for several tissues and organs, including the brain. An unexpected result was finding that after a single i.v. bolus administration of HAEE to rabbits and rats, an atypical (non-canonical) type of pharmacokinetic dependence was observed for this kind of experiment, i.e., the presence of a pronounced maximum of the peptide concentration at the beginning of the pharmacokinetic curve, 4 min after its administration (Figure 1A,B and Figure 2A,B). Moreover, in experiments with rats (Figure 2A,B), this effect was reproduced for all tested doses of HAEE: 50, 300 and 900 μg/kg. At a subsequent part of the pharmacokinetic curve (starting from 4 min and further after the introduction of the peptide), the resulting dependence was well-approximated by the bi-exponential decay equation (Equation (1)). This equation is used to mathematically describe open, two-compartment pharmacokinetics with elimination of the substance from the central compartment (Figure 1D). In this case, the first exponent (characterized by macro-constants A and α) mainly reflects the process of substance distribution between the central and peripheral compartments, and the second exponent (characterized by macro-constants B and β) reflects the process of substance elimination from the central compartment. The use of a simpler one—compartment model, described by a mono-exponential decay equation, demonstrates a lower approximation estimate. Thus, starting from minute 4 (4–120 min for rabbits and 4–30 min for rats), the pharmacokinetic behavior of the HAEE peptide fit well within the framework of the classical two-compartment pharmacokinetic model. 

The non-canonical shape of pharmacokinetic dependence that we discovered, in the time range of 0–4 min, can be explained by the intervention in the distribution of HAEE in the blood–tissue system of the following two processes: (1) binding of HAEE to acceptor sites located on the endothelium on the side of the lumen of blood vessels, and (2) binding of HAEE with blood plasma proteins. In addition, it is known that pharmacological substances with acidic properties have high affinity for blood plasma proteins [43]. It can be conjectured that immediately after the introduction of HAEE into the central bloodstream, it instantly binds to numerous affinity acceptor sites located on the surface of the vascular endothelium, which is reflected in its “underestimated” amount in blood plasma. In addition, the peptide binds to plasma proteins, but less readily than to acceptors located on the endothelium. Further, over time, there is a more uniform distribution of the HAEE pool associated with the endothelium, along the entire surface of vascular endothelium. At the same time, there is a redistribution of the peptide between its two pools, the pool of the peptide bound to the endothelium and the pool of the peptide bound to plasma proteins, leading to an increase in the amount of peptide bound to plasma proteins and to a decrease in the amount of the peptide bound to the endothelium. In terms of the pharmacokinetic dependence, this is expressed in the observed increasing concentration of HAEE in the blood plasma up to 4 min after i.v. administration. 

The increase in the amount of HAEE bound to plasma proteins, most likely to serum albumin, can be attributed to the process of gradual filling of free binding sites on acceptor proteins with the peptide. It is possible that binding of the HAEE peptide to plasma proteins proceeds kinetically somewhat more slowly than the binding of the peptide to endothelial acceptors. Thus, the dynamic equilibrium between the pools of free and plasma protein-bound HAEE peptides is established more slowly than the dynamic equilibrium between the pools of free and endothelium-bound HAEE peptides. At the same time, both pools of HAEE acceptors can be characterized by high capacity, since an increase in the peptide dose did not significantly affect the appearance of the noncanonical portion of the pharmacokinetic curve (Figure 2A,B). Finally, at the time point of 4 min after the administration of the peptide, we observed establishment of a dynamic equilibrium between three conditional pools of peptide: (1) free (not bound to proteins) in blood plasma, (2) bound to blood plasma proteins and (3) bound to acceptor sites located on the endothelium.

Based on the analysis of data from the experiments on rabbits and rats (Figure 1 and Figure 2), the main pharmacokinetic parameters were calculated for a single i.v. bolus administration of the HAEE peptide (Table 2). The calculated half-lives (T_1/2(el)_) of the HAEE peptide from rabbit and rat blood were 20 and 22 min, respectively. Considering that for most peptides the half-life usually does not exceed several minutes, these data indicate a relatively high stability of the HAEE peptide in the organism. Most likely, this is due to the presence of protected *N*- and *C*-termini [20]. Confirmation of the high resistance of such a modified peptide to proteolytic hydrolysis was provided by the results we obtained in in vitro experiments (Section 3.1), indicating that the peptide remains unchanged in blood plasma. In this regard, excretion through the kidney or liver seems to be the most likely route of excretion of the peptide from the body. It is of interest that analysis of the biodistribution of the peptide in mouse tissues (given below) revealed its extremely high concentration in the kidney (Figure 3A). It is also important to note that for a wide range of HAEE doses (50–900 μg/kg) there is a clear linearity of its pharmacokinetics in blood (Figure 2C). We did not find significant deviations in the pharmacokinetic behavior of the HAEE peptide after its single bolus administration to animals received daily for 28 days at a dose of 300 μg/kg, which indicated the absence of cumulative and “depleting” effects of chronic peptide administration on its pharmacokinetics (Table 2, Figure 2D). It should also be noted that the HAEE peptide, when administered repeatedly to rats, did not cause visible toxic effects and abnormalities in animal behavior.

The biodistribution of the HAEE peptide was studied in mice (Figure 3A–C). In this case, intraperitoneal (i.p.) administration of the peptide was used. The analysis of the HAEE tissue bioavailability showed that its highest amount was found in kidneys, where its concentration in the entire studied time range exceeded the concentration in the blood (Table 3). A significantly lower amount of the peptide, but comparable with blood, was found in the liver and heart, and the lowest in the omentum and the brain. The fact that the highest amount of HAEE was found in kidneys is in good agreement with the results of a large-scale study on the distribution of more than 200 pharmacological substances in the body [43], which showed that substances with acidic properties tend to exhibit high tropism to the kidneys. 

Noteworthy was the unusual form of the pharmacokinetic curves for HAEE obtained for different tissues (Figure 3A–C). As with the one-time bolus i.v. administration, it turned out to be atypical. The classic (canonical) pharmacokinetic curve for a single bolus i.p. administration is a monomodal bell-shaped curve with a maximum, on which two characteristic sections are manifested: (1) a section with an increase in the concentration of substance that then reaches a maximum, which corresponds to the time range 0–T_max_ and characterizes the process of absorption of a substance into the blood from the peritoneal fluid, and its distribution in tissues; and (2) a section with a monotonic decrease in the concentration of a substance, described by a mono-exponential dependence, which corresponds to the time range T_max–__∞_ and characterizes the process of excretion of a substance from the body. In the case of HAEE with i.p. administration, we observed a pronounced bimodal relationship for all tested tissues, with the possible exception of the omentum (Figure 3A–C). In this case, the first mode corresponded to the time range of about 4 min after HAEE administration and the second to the time range of about 10–15 min after its administration. The ratio of values of modal concentrations differs for different tissues: for blood the value of the first mode is much higher than the second, for kidneys there is an inverse relation, and for the heart and liver they are comparable. Finally, a slightly higher second mode was observed for the brain.

We assumed that the first mode reflected the classical type of distribution of the peptide coming from the blood to peripheral tissues (the first phase of distribution) and that the second, atypical mode was due to the presence of interstitial distribution of the peptide (the second phase of distribution) between easily accessible regions of peptide deposition (capillary blood, lymph, interstitial fluid, the outer surface of cells) and hard-to-reach locations for deposition (for example, the intracellular space) that have a greater ability to keep the peptide unchanged. Moreover, easily accessible regions of peptide deposition are characterized by a rapid establishment of a dynamic equilibrium with blood, while those that are difficult to access are characterized by slow one. An important factor affecting the magnitude of the modes is ostensibly the pH of the liquid medium in tissues and the density of affinity acceptors capable of binding the peptide. It should also be noted that the analysis of the pharmacokinetic data obtained for the case of i.v. administration did not reveal the second phase of distribution, apparently because it was “masked” by a more powerful first phase of distribution of the peptide between blood and the peripheral tissues. 

Changes in the HAEE concentration in the brain were also characterized by a bimodal relationship with a higher second mode (Figure 3C). These data indicate that the HAEE peptide accumulates in the brain against the background of a decrease in its concentration in the blood (Figure 3C). For about 10 min (from 4 to 15 min after i.p. injection), the concentration of HAEE in the brain is kept at a stable level, which possibly allows the peptide to effectively interact with Aβ aggregates on the surface of neuronal cells and ultimately cause their dissociation and destruction. 

The blood–brain barrier (BBB) is a major obstacle to drug delivery into the central nervous system (CNS), in particular for peptides and proteins. Unfortunately, none of the known drug prototypes that have an anti-amyloid effect in vitro and in animal models of AD have been shown to be effective in clinical trials [44]. One of the explanations for this inefficiency is the inability of the developed drugs to penetrate the BBB [45]. Therefore, the mechanism of HAEE transport from blood to brain parenchyma is an important issue [46].

We have shown by molecular modeling that HAEE might represent one of the low-density lipoprotein-related protein-1 (LRP1) targeting peptides. Initially, we noticed the similarity of the motif of charged amino acid residues (ion-complementary motif) of HAEE with the KTEE site in the C-terminal region of the Angiopep-2 peptide. It is known that Angiopep-2 penetrates the BBB largely due to receptor-mediated transcytosis (RMT), where LRP1 acts as a receptor [9]. It is also known that Angiopep-2 binds to the beta-propeller B1 domain of this receptor, but the exact binding site of Angiopep-2 is unknown [10]. We found that the B1 domain of the LRP-1 receptor contains an HHVE region, which is a mirror image of the EVHH sequence of beta-amyloid, and it is this HHVE region that is the most likely site for Angiopep-2 binding through its KTEE site (Figure 6). Thus, we could conjecture that HAEE would also specifically bind to the HHVE site. Indeed, molecular modeling showed that the site of the most likely binding of the HAEE peptide in the model structures of the B1 domains of murine and human LRP1 receptors is the HHVE region of these receptors (Figure 4 and Figure 5). The data obtained in silico indicated that HAEE passes through the BBB via the receptor-mediated transcytosis mechanism, which is the optimal route for drug delivery (in this case, HAEE) to the central nervous system. 

In combination with the previously established anti-amyloid effects of HAEE peptide in a mouse model [8] and in vitro experiments [5], the pharmacokinetic parameters of the HAEE peptide determined in this work and the obtained model for the mechanism of possible passage of this peptide through the BBB allow us to consider HAEE as a promising pathogenetic (disease-modifying) drug for the treatment of Alzheimer’s disease.

## Figures and Tables

**Figure 1 biomolecules-11-00909-f001:**
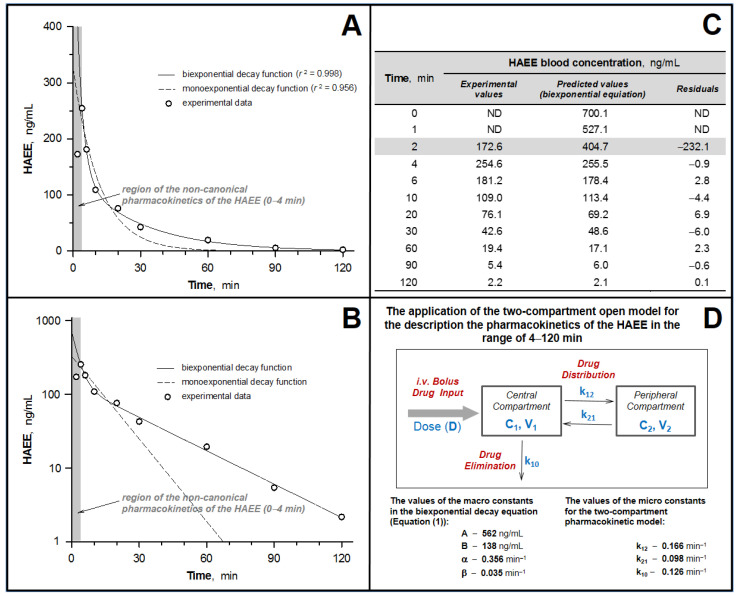
Pharmacokinetics of HAEE in rabbit blood after a single intravenous (i.v.) bolus injection at a dose of 120 μg/kg. (**A**,**B**). Approximation of experimental data by equations of mono- and bi-exponential decay functions. Data are presented in straight (**A**) and semi-logarithmic (**B**) coordinates. (**C**). Deviation of experimental data from the results predicted by the biexponential decay function in the time range 0–4 min. (**D**). Calculation of pharmacokinetic parameters in an open two-compartment model for i.v. bolus administration in the time range 4–120 min.

**Figure 2 biomolecules-11-00909-f002:**
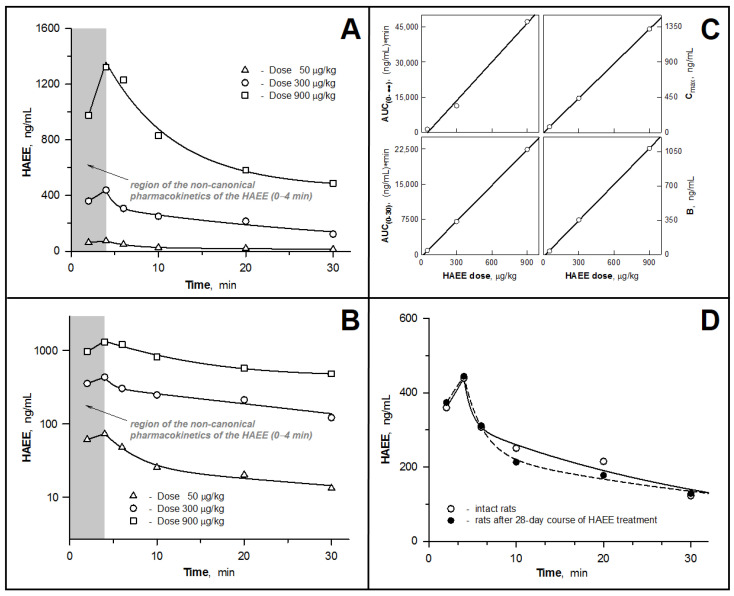
Pharmacokinetics of HAEE in rat blood after a single intravenous bolus injection at doses of 50, 300 and 900 µg/kg. (**A**,**B**). Effect of HAEE dose on HAEE pharmacokinetics. Data are presented in straight (**A**) and semi-logarithmic (**B**) coordinates. (**C**). Verification of the linearity of HAEE pharmacokinetics versus the peptide dose. (**D**). Analysis of the effect of long-term course administration of HAEE on the pharmacokinetics of the peptide.

**Figure 3 biomolecules-11-00909-f003:**
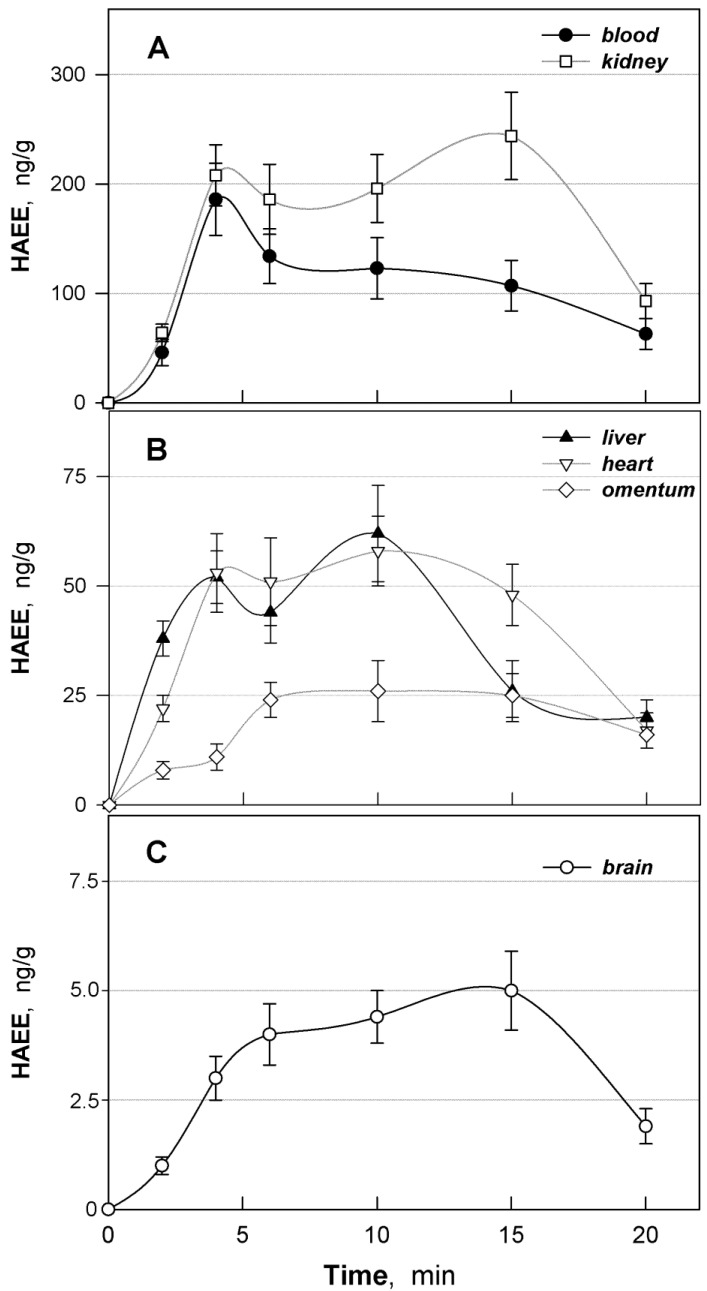
Distribution of HAEE between organs and tissues of mice after intraperitoneal bolus injection at a dose of 300 μg/kg. (**A**). Quantity of HAEE in blood and kidneys; (**B**). quantity of HAEE in the liver, heart, and omentum; (**C**). quantity of HAEE in the brain.

**Figure 4 biomolecules-11-00909-f004:**
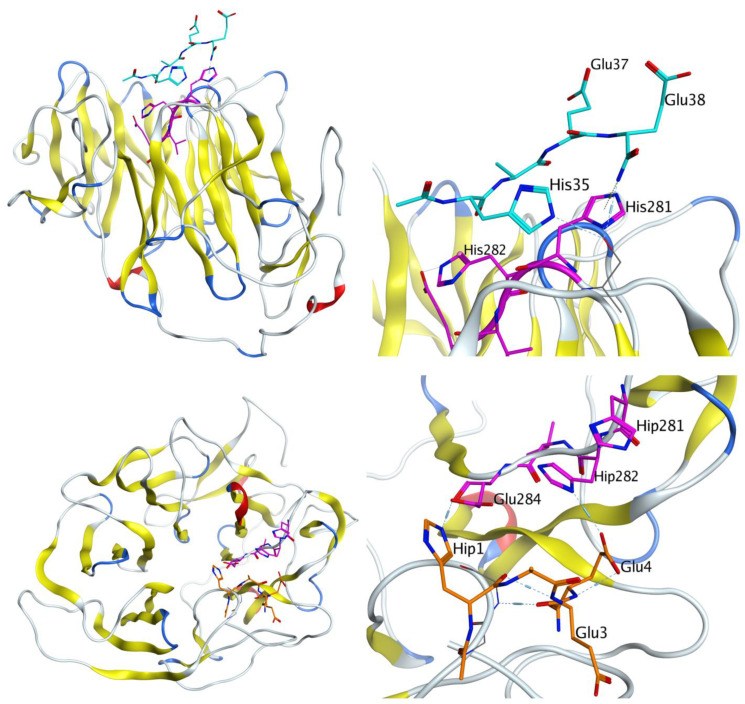
Interaction of the HAEE tetrapeptide (marked blue for the unprotonated system, marked orange for the protonated system) with the 281-HHVE-284 site (marked pink) of LRP1 after 50 ns (top) and 20 ns (bottom) of MD equilibration.

**Figure 5 biomolecules-11-00909-f005:**
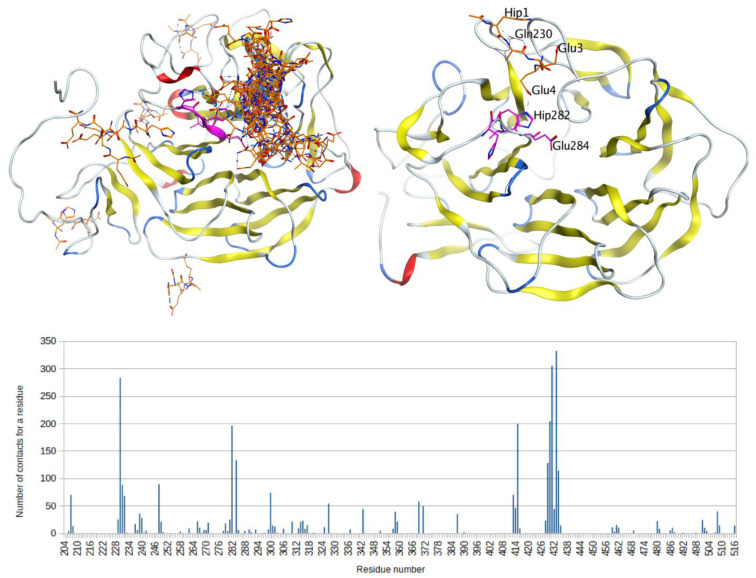
Results of global docking of the protonated tetrapeptide HAEE (marked orange) to LRP1 (left), and the structure of the model with the initial location of the tetrapeptide relative to the HHVE site after 50 ns MD (right). The HHVE site is marked pink. The bar graph shows the total number of contacts for all docking models for each amino acid residue of LRP1.

**Figure 6 biomolecules-11-00909-f006:**
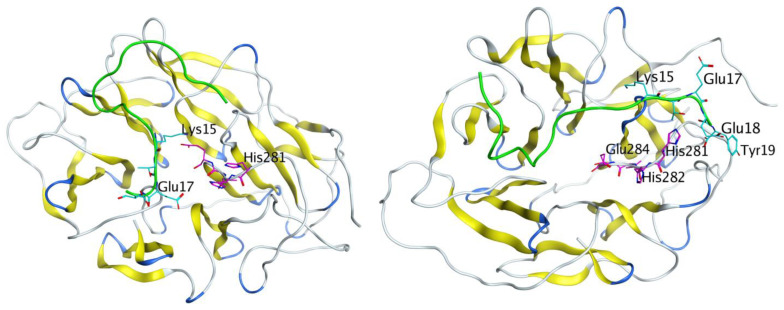
Interaction of angiopep-2 (marked green) with mouse LRP1 after 30 ns (left) and 50 ns of MD (right). The KTEE site is marked blue and the HHVE site in LRP1 is marked pink.

**Table 1 biomolecules-11-00909-t001:** Experimental groups.

Experimental Group	Animal Species	*n*	Study Type	Dose of HAEE, μg/kg	HAEE Route of Administration
1	*Rabbit*	3	Pharmacokinetics in blood	120	i.v.
2	*Rat*	6	Pharmacokinetics in blood (dose dependence)	50	i.v.
3	*Rat*	6		300	i.v.
4	*Rat*	6		900	i.v.
5	*Rat*	6	Pharmacokinetics in blood (effect of chronic administration ^a^)	300	i.v.
6	*Mouse*	36	Tissue distribution	300	i.p.

^a^ Rats of experimental group 5 were i.p. injected for 28 days prior to pharmacokinetic studies with unlabeled HAEE at a dose of 300 μg/kg/day.

**Table 2 biomolecules-11-00909-t002:** Calculated values of the main pharmacokinetic (PK) parameters of HAEE in the blood of test animals after a single intravenous bolus injection of the peptide.

**PK Parameter ^a^**	**Units**	**Dose of Peptide**, μg/kg				
		*Rabbit*	*Rat*			
		120	50	300	300 ^b^	900
D	ng	440,000	18,000	105,000	105,000	315,000
C_max_	ng/mL	255	73.9	439	444	1322
T_max_	min	4	4	4	4	4
AUC_(0–__∞)_	(ng/mL) × min	4421	1383	11,340	12,137	47,133
Cl_T_	mL/min	100	13.0	9.3	8.7	6.7
MRT	min	29	33	32	42	36
K_el_	min^−1^	0.035	0.031	0.031	0.024	0.028
T_1/2(el)_	min	20	23	22	29	25
V_d(c)_	mL	862	107	122	111	108
V_d(β)_	mL	2860	425	299	362	237
V_d(extrap)_	mL	3194	504	296	382	291

^a^ D indicates dose of a pharmacological substance; C_max_ is the maximum concentration of a substance in the blood; T_max_ is the time at which the maximum concentration of the substance in the blood is reached; AUC_(0–__∞)_ is the area under the pharmacokinetic curve from the moment of its introduction to complete elimination from the body; Cl_T_ is total body clearance; MRT is the residence time of a substance in the body; K_el_ is the constant of the rate of excretion of the substance from the central compartment in the elimination phase; T_1/2(el)_ is the half-life of the substance in the central compartment in the elimination phase; V_d(c)_ is the volume of distribution of the substance in the central compartment when C_max_ is reached; V_d(β)_ is the volume of distribution of the substance in the elimination phase; V_d(extrap)_ is the volume of distribution of a substance upon reaching a concentration equal to the macro-constant B in the bi-exponential decay equation (Equation (1)). ^b^ The pharmacokinetic study was performed after a 28 day course of administration of unlabeled HAEE at a dose of 300 μg/kg/day.

**Table 3 biomolecules-11-00909-t003:** Evaluation of HAEE bioavailability (*f_T_*) for several organs/tissues of mice after intraperitoneal bolus administration at a dose of 300 μg/kg.

Organ/Tissue	^a^ AUC_(0–20)_, (ng/g) × min	*f_T_*	*f_T_* × 100%
Blood	2110	1.000	100.0
Kidney	3437	1.628	162.8
Liver	777	0.368	36.8
Heart	845	0.400	40.0
Omentum	395	0.187	18.7
Brain	70	0.033	3.3

^a^ AUC_(0–20)_ is the area under the pharmacokinetic curve in the time period 0–20 min. Bioavailability values were calculated as: *f_T_* = AUC_(0–20)_ tissue/AUC_(0–20)_ blood.

## Data Availability

Data are available upon request.

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
