# Peer review of "Pharmacokinetics and Molecular Modeling Indicate nAChRα4-Derived Peptide HAEE Goes through the Blood–Brain Barrier"

_biomolecules, 2021, doi:10.3390/biom11060909_

Round 1

Reviewer 1 Report

It is an interesting and well documented study addressing the pharmacokinetics of HAEE peptide administered in vivo. The idea of using this peptide to prevent amyloid-beta aggregation in the brain is quite relevant. A principal question, which is not even mentioned in the manuscript, is that the main target of amyloid-beta in the brain is considered to be alpha7 nAChR (Parri HR, Dineley KT. Nicotinic acetylcholine receptor interaction with β-amyloid: molecular, cellular, and physiological consequences. Curr Alzheimer Res. 2010; 7( 1): 27-39; Wang HY, Lee DH, D'Andrea MR, Peterson PA, Shank RP, Reitz AB. β-Amyloid 1–42 binds to α7 nicotinic acetylcholine receptor with high affinity. Implications for Alzheimer's disease pathology. J Biol Chem. 2000; 275:5626–5632). How do the authors coordinate their results/ideas with these data?

The English language of the manuscript requires improvement.

Author Response

Thank you very much for your valuable comments. Below are our responses.

Point 1: QUESTION It is an interesting and well documented study addressing the pharmacokinetics of HAEE peptide administered in vivo. The idea of using this peptide to prevent amyloid-beta aggregation in the brain is quite relevant. A principal question, which is not even mentioned in the manuscript, is that the main target of amyloid-beta in the brain is considered to be alpha7 nAChR (Parri HR, Dineley KT. Nicotinic acetylcholine receptor interaction with β-amyloid: molecular, cellular, and physiological consequences. Curr Alzheimer Res. 2010; 7( 1): 27-39; Wang HY, Lee DH, D'Andrea MR, Peterson PA, Shank RP, Reitz AB. β-Amyloid 1–42 binds to α7 nicotinic acetylcholine receptor with high affinity. Implications for Alzheimer's disease pathology. J Biol Chem. 2000; 275:5626–5632). How do the authors coordinate their results/ideas with these data?

Response 1: We agree with the opinion of the Reviewer and in the revised manuscript, in the Discussion section (lines 449-464), we added the following paragraph, including data on the role of beta-amyloid interactions with the alpha7 nicotinic acetylcholine receptor:

It can be assumed that Aβ deposition induces degeneration of cholinergic terminals [25], especially at the locations of α4β2 nAChRs [26,27] and a7 nAChRs [28]. Many studies support the notion that Aβ can physically interact with α4β2 nAChRs and α7 nAChRs in various model systems [29-32]. Since Aβ accumulates in the brain regions enriched in α4β2 nAChRs and α7 nAChRs, the selective vulnerability of the hippocampus to Aβ toxicity can be associated with the high-affinity interaction between Aβ and these nAChRs [33-36]. The Ab - α4β2 nAChR interaction interface is formed by sites 11-EVHH-14 and 35-HAEE-38 of Ab and α4 subunit of α4β2 nAChR, respectively [5]. The 11-EVHH-14 region of Aβ also plays a critical role in Aβ binding to a7 nAChRs, however, the exact interface of the Ab - α7 nAChR complex is unknown [4,6,29,37,38]. The 11-EVHH-14 region has a relatively rigid backbone conformation in soluble Aβ monomers [39,40] and zinc-bound dimers [41]. This site corresponds to the b-strand b2 from the N-terminal arch of the Aβ amyloid fibrils purified from Alzheimer's brain tissue and is solvent exposed and accessible for interactions with external molecules [42]. Thus, molecular agents binding to the 11-EVHH-14 region of Aβ can modulate interactions between Aβ (in soluble or aggregated states) and α4β2- and α7-containing nAChRs.

Point 2: The English language of the manuscript requires improvement.

Response 2: After consulting with a native speaker, we made all the necessary corrections and improvements (highlighted in yellow) to the revised manuscript.

Reviewer 2 Report

Zolotarev et al. describe the pharmacokinetics of an anti-amyloid tetrapeptide on three types of laboratory animals and model the interactions peptide-transport protein as a putative mechanism for BBB permeability.

Minor revisions:

  1. Combine Tables 2 and 3.
  2. Improve the visibility of residue names in Figures 4 - 6.

Author Response

Thank you for your attention to our work. Below are our responses.

Point 1: Combine Tables 2 and 3. 

 Response 1: Done.

Point 2: Improve the visibility of residue names in Figures 4 - 6

Response 2: Done.
